# Defect free strain relaxation of microcrystals on mesoporous patterned silicon

Alexandre Heintz [1,2] ✉, Bouraoui Ilahi [1,2], Alexandre Pofelski[3], Gianluigi Botton [3,4], Gilles Patriarche[5], Andrea Barzaghi [6], Simon Fafard[1,2], Richard Arès [1,2], Giovanni Isella [6] & Abderraouf Boucherif [1,2] ✉

A perfectly compliant substrate would allow the monolithic integration of high-quality semiconductor materials such as Ge and III-V on Silicon (Si) substrate, enabling novel functionalities on the well-established low-cost Si technology platform. Here, we demonstrate a compliant Si substrate allowing defect-free epitaxial growth of lattice mismatched materials. The method is based on the deep patterning of the Si substrate to form micrometer-scale pillars and subsequent electrochemical porosification. The investigation of the epitaxial Ge crystalline quality by X-ray diffraction, transmission electron microscopy and etch-pits counting demonstrates the full elastic relaxation of defect-free microcrystals. The achievement of dislocation free heteroepitaxy relies on the interplay between elastic deformation of the porous micropillars, set under stress by the lattice mismatch between Ge and Si, and on the diffusion of Ge into the mesoporous patterned substrate attenuating the mismatch strain at the Ge/Si interface.

Over the past decades, semiconductor technology has become essential for automotive, biomedical, sensing, and environmental monitoring applications. Devices such as light-emitting diodes, photodetectors, lasers and solar cell are ubiquitous in consumer, industrial,and scientific appliances[1–6]. These complex devices rely on epitaxial growth ensuring high crystalline quality, abrupt interface, accurate alloy composition and doping level. However, crystal growth remains sensitive to the difference in lattice parameter between the epilayer and the substrate as well as the difference in thermal expansion coefficients[7–9]. Depending on the lattice mismatch, strained epitaxial material fitting the substrate structure, can be grown pseudomorphically ensuring coherent strain distribution[10] up to a critical thickness when plastic relaxation and defect creation appear as an elastic energy relaxation mechanism[11]. Beyond this critical thickness, the epitaxial layer will release the strain energy giving rise to defect nucleation such as misfit and threading dislocations, which are

detrimental to device performance and must therefore be avoided or at least minimized[12,13]. In heteroepitaxy, the difference in lattice parameter between the substrate and the epitaxial layer require some kind of mitigation in order to reach application quality material. Without such mitigation, the variety of suitable substrates that can ensure high quality epitaxy is limited compared to the diversity of materials and alloys required for the synthesis of high performances devices.

To tackle this problem, different attempts involving the use of standard substrates have been proposed over the past decades to reduce the threading dislocation density (TDD)[14–20], which is the main source of detrimental effects on the epilayer. Metamorphic growth involving several microns thick buffer layer either by compositional grading[16–18] or several high temperature annealing cycles[19], has particularly achieved attractive results, and has succeeded in decreasing the TDD down to a threshold of $10^6$ TD/cm$^2$. Other alternative methods consider patterned substrates as a mean to reduce the defects[21–23].

[1]Institut Interdisciplinaire d'Innovation Technologique (3IT), Université de Sherbrooke, 3000 Boulevard Université, Sherbrooke, QC J1K OA5, Canada. [2]Laboratoire Nanotechnologies Nanosystèmes (LN2) —CNRS UMI-3463, Institut Interdisciplinaire d'Innovation Technologique (3IT), Université de Sherbrooke, 3000 Boulevard Université, Sherbrooke, QC J1K OA5, Canada. [3]Department of Materials Science and Engineering, McMaster University, Hamilton, ON L8S 4M1, Canada. [4]Canadian Light Source, 44 Innovation Boulevard, Saskatoon, SK S7N 2V3, Canada. [5]Centre de Nanosciences et de Nanotechnologies – C2N, CNRS, Univ. Paris-Sud, Université Paris-Saclay, 91120 Palaiseau, France. [6]L-NESS and Dipartimento di Fisica, Politecnico di Milano, Via Anzani 42, I-22100 Como, Italy. ✉e-mail: alexandre.heintz@usherbrooke.ca; abderraouf.boucherif@usherbrooke.ca

Accordingly, three-dimensional (3D) growth of Ge and/or SiGe on deeply patterned Si substrate has been found efficient for TDs elimination by allowing them to propagate towards the free surface of the pattern features sidewalls[24–26]. However, these methods still require thick buffer layer with relatively high defect density, mostly around $10^6$ TD/cm², associated with well pronounced misfit dislocations (MF) network, which precludes the synthesis of devices demanding very thin active layers close to the interface. Recent study has shown that nanovoid-based virtual substrate (NVS) can be used to decrease the TDD down to $10^4$ TD/cm², but still limited to Ge/Si system[27]. Moreover, metamorphic growth implies long growth times to create the thick transition layer, effectively increasing the time and cost of the desired structure.

Back in the nineties, Lo and Teng proposed a method based on a theoretical model called "compliant substrate," which is supposed to create the conditions for defect-free heteroepitaxy[28,29]. This method is based in the idea of reducing the substrate's effective thickness, so that the compliant substrate accommodates a large part of the strain, increasing the critical thickness of the epilayer and allowing a pseudomorphic growth of thicker defect-free layer. Several studies have been performed to experimentally demonstrate this theory, for instance the development of free-standing nano-membranes[30] and ion-implantation[31]. This later can be used to reduce the difference in lattice parameter, while thin membrane allows the epilayer to relax to its natural lattice constant. The membrane can be then transferred to a host substrate[32]. Despite promising results shown by these methods, achieving an effective and practically useful compliant substrate remains a challenge. Indeed, the very thin nature of such membranes engenders handling difficulties during microfabrication and layer transfer processes regarding surface contamination and mechanical stability of the final device. Other methods have also been proposed in the literature to reach compliance[33,34], but still present limitations regarding their scalability.

Van Der Waals heteroepitaxy on graphene has recently generated increasing interest[35]. With this method the strain is taken up by the graphene intermediate layer, since its deformation energy is lower than the one required to form a defect due to plastic relaxation. Since the Van der Waals' bonds are very weak, growth on graphene also presents benefits regarding easier layer transfers. However this method requires the graphene transfer, which is difficult to implement at industrial scale[36].

Porous silicon is a promising material to reach such a compliance. Indeed, its mechanical properties can be tuned depending on the porosity, leading to an elastic material with low Young's modulus while remaining crystalline[37,38]. Several studies have been performed on the growth of GaAs, SiGe or Ge on mesoporous silicon[39–41]. However, these studies did not highlight the compliant properties of standard porous silicon substrate. Only a slight improvement of crystalline quality has been noticed. Ge deposition on planar porous Si substrate has been reported, the TDD has been shown to be reduced down to $2.4 \times 10^7$ cm² after annealing steps[42,43]. Free standing graphene mesoporous Si membrane has already been proposed as complaint substrate for the growth of GaN with high potentiality to accommodate the strain energy during epitaxy[44]. Nevertheless, the effectiveness of conventional porous silicon as a suitable compliant substrate is limited by the reorganization of the porous structure during the epitaxial process[45–47] involving high temperature, the lattice accommodation between the substrate and the epilayer or even by the brittleness of the porous silicon membrane. None of the methods employed, such as porous template, patterned substrate or even SiGe graded layer can account alone for complete suppression of TD.

In this work, we propose a fully compliant Si substrate as a practical way towards the long-standing goal of defect free heteroepitaxy of lattice mismatched materials on the industry-standard silicon substrates. The proposed solution paves the way towards monolithic integration of wide range of optoelectronic devices with advanced applications and functionalities through band engineering on Si substrate. The method relies on the deep patterning of the Si substrate to form micrometer scale pillars and subsequent electrochemical porosification. Considering the Ge epitaxy on Si substrate as a case study, the tower morphology of the Si pillars allows full elastic relaxation of the thermal strain[48] whereas, the porosification reduces the pillars Young's modulus allowing easy deformation to accommodate the lattice mismatch strain. The results demonstrate the full compliance of the silicon substrate revealing unprecedented dislocation-free Ge microcrystals regardless the deposited thickness. Our finding paves the way to achieve high quality germanium for active photonic devices on Si platform.

## Results and discussion

### Ge growth on mesoporous Si patterned substrate

Typical cross section SEM micrograph of the compliant Si substrate is shown by Fig. 1. The substrate is formed by deep patterning and porosification.

SEM image of Bosch process deeply patterned p-type Si (001) wafers (10–20 mOhm.cm) with ordered square-based $5 \times 5$ cm² arrays of Si pillars separated by 1 µm trenches used as substrates is shown by the Fig. 1a.

The obtained Si pillars were anodized (Fig. 1b) in O-ring electrochemical cell with an electrolyte composed of 1:3 volume ratio of HF (49%) and anhydrous ethanol and a 50 mA/cm² current density to form a 2 µm-thick mesoporous Si pillars with $70 \pm 5$% porosity. The electrochemical etching of silicon pillars leads to the formation of a dendritic morphology perpendicular to each free surface exposed by the patterned substrate (Supplementary Fig. S1).

A 200 nm thick Ge buffer layer has been first deposited at 200 °C prior to the growth of 2 µm-thick Ge layers by chemical beam epitaxy at 500 °C (TC) using a solid source Ge. The aim of the low temperature buffer layer is to close the pores, defining then a suitable flat surface and to minimize the Ge diffusion into the porous Si pillars. An identical growth procedure has been used to deposit Ge on a patterned substrate which did not underwent the porosification procedure. Due to the patterning of the substrate, 3-dimensional Ge microcrystals are obtained on both Si pillar (SiP) and porosified Si pillars (PSiP) (Fig. 1c, d). It has been previously shown[24] that the growth of Ge at low deposition temperature enhances the vertical growth of Ge crystal on the Si pillars and, optimizing the deposition conditions it is possible to expel threading dislocations at the lateral sidewalls of the Ge microcrystal.

### Defect density in Ge microcrystals

A full elastic strain accommodation should be accompanied by the absence of TDs. To validate this, etch-pit (EP) counting has been performed on both Ge epitaxial material deposited on SiP and PSiP. For this purpose, samples were immersed in a solution of two volumetric parts 49 wt% HF and 1 part 0.1 M $K_2Cr_2O_7$, where mixed and screw dislocations in the Ge material get selectively etched allowing their quantification by using plan view SEM observations. The average defect density has been extracted from different Ge/SiP and Ge/PSiP top-view SEM images. As shown by the Fig. 2a, relatively high TDD around $5 \times 10^8$ cm² is found to reach the surface for Ge grown on SiP. Meanwhile, for Ge grown on PSiP, the surface appears completely free of pits implying that no TD dislocation reaches the surface (Fig. 2b). Bending of the towers can be observed in Fig. 2b due to the flexibility of the porous silicon pillars. To assess the impact of intermediate porosity on the TDD, Etch-pit counting on Ge microcrystals grown on 50% PSiP (Supplementary Fig. 4) revealed that the TDD decreases down to $2.5 \times 10^8$ cm². The decrease of the threading dislocation density confirms that the crystalline quality of the Ge microcrystal's gets progressively improved with increasing the PSiP porosity.

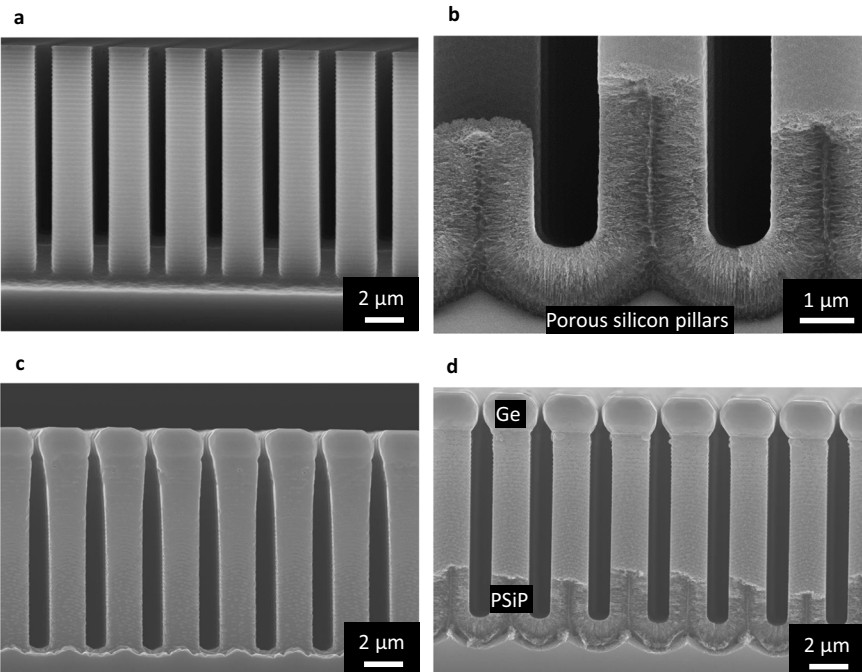

**Fig. 1 | Compliant substrate realized using patterned Si(001) porous substrate.** **a** Cross-section SEM image of 10 μm tall deeply patterned silicon wafer using Bosch process. **b** Cross-section SEM image of the bottom part of anodized silicon pillars, the upper part of the porous Si pillars cannot be observed by SEM due to the screening effect coming from the Ge deposition on the lateral surface and the cleavage, however the TEM analysis reported in the Supplementary Material shows the reorganized porous structure over the entire Si pillar. **c, d** 2 μm tall self-limited Ge microcrystals grown at 500 °C by chemical beam epitaxy using Ge solid source on SiP (**c**) and on PSiP (70%) (**d**).

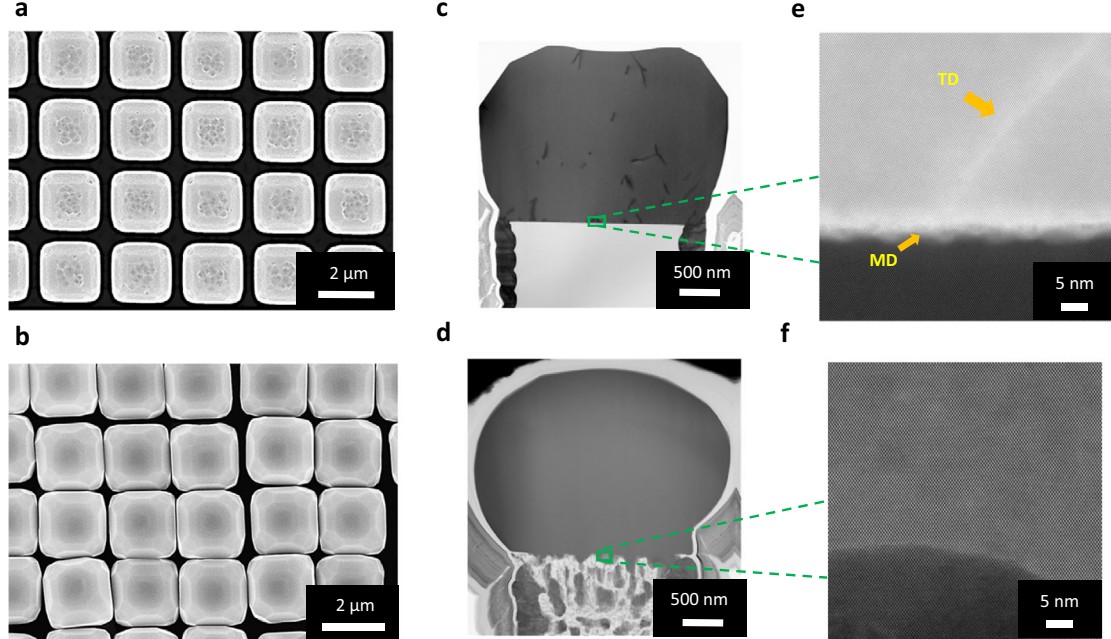

**Fig. 2 | STEM characterization of Ge microcrystals along the <110> zone axis.** **a, b** Top-view SEM image of Ge/SiP (**a**) and Ge/PSiP(70%) (**b**) after Etch-pit (EP) method. c,d, Low magnification BF-TEM images of the Ge/SiP reference substrate (**c**) and of the defect-free germanium grow on Porous silicon pillar (PSiP) (**d**). **e** High-resolution STEM image of the Ge/Si interface with threading dislocations and misfit network. **f** Atomic resolution STEM image of the Ge/PSiP interface.

The facet formation has been observed in all the investigated structures regardless the substrate preparation procedure as shown by the Top view SEM images (Supplementary Fig. 4). In the meanwhile, the pillars porosification is found to induce faceting morphological changes. The difference in morphology can be attributed to the lack of a unique/dominant plan due to the porous template. The porous silicon pillars exhibit crystallites, each of them constitutes a nucleation site for Ge crystal resulting in a reduced area of <001>facet in favor of increasing that in the other crystallographic orientations. In some cases (see e.g., Fig. 2c) the microcrystals feature a rounded morphology. This is indeed expected in group IV crystallites at thermodynamic equilibrium[49].

This result constitutes a first confirmation that the PSiP concept succeeds in accommodating the lattice mismatch strain. Obtaining such defect-fee microcrystals on a large-scale area, as shown by EP counting, by combining Bosch process, porosification and epitaxial growth, indicates that our method could potentially be scalable for industrial production.

To gain more insights on the structural properties of the Ge microcrystals, and more precisely the interface and in-depth distribution of the dislocations, TEM cross-sectional observation of Ge epitaxial material on SiP and on PSiP have been performed (Fig. 2c, d).

Indeed, as shown by SEM (Fig. 2a) and TEM (Fig. 2c) micrographs, the Ge grows on SiP with multiple facets due to the pillars tower morphology[24]. Threading dislocations that are not parallel to the growth direction are expelled to the sidewall of the pillars. It has already been shown[25] that it is possible to completely expel TDs from the top part of the microcrystal, however, this requires the combination of suitable growth conditions and pillar widths, together with a relatively large microcrystal thickness[24]. As we can deduct from the TEM observations (Fig. 2c) of the reference sample, the Ge top surface is partly faceted with the presence of (001) central surface, surrounded by (113) facets. Since growth dislocation propagate in a perpendicular direction of the facets, those vertical dislocations can reach the surface. As expected, defects appear in Ge microcrystal on SiP due to lattice mismatch. However, in the case of Ge grown on the porous structure (Fig. 2d), no threading nor misfit dislocations appeared in the Ge epitaxial material.

These widely sought properties can be exploited to achieve Ge based photonic and optoelectronic devices that require very thin thickness[50]. This constitutes a direct proof of porous Si pillars properties being able to accommodate the mismatch strain. Selected area electron diffraction (SAED) images (Supplementary Fig. 5e, f) highlight a monocrystalline Ge microcrystal, while the porous structure presents an enlargement of the dots and a deformation, which indicate a porous crystallites deformation.

EDX observations in Supplementary Fig. 6, revealed the penetration of Ge into the porous structure due to high specific surface. On the other hand, a graded interdiffusion of Ge and Si can be observed. Due to the Ge diffusion length, the Ge concentration in the resulting SiGe graded structure, increases near the interface and the sidewalls (Supplementary Fig. 7). This SiGe synthesis can also contribute to the dots deformation in SAED patterns.

## Crystal quality analysis

In order to assess the crystalline quality and residual strain, high resolution x-ray diffraction, coupled with reciprocal space mapping around the symmetrical Si(004) and asymmetrical Si(224) reflections for both heterostructures grown on SiP and PSiP (70%) substrates were performed, Fig. 3.

As can be observed on the coupled scans (Fig. 3a, b), the full width at half maximum (FWHM) of the Ge peak grown on the porous structure is narrower than the one obtained with the reference on SiP, which confirm the improvement of the crystalline quality using such compliant substrate. Additionally, in case of Ge growth on PSiP an asymmetric broadening occurs for both Ge and Si diffraction peaks (Supplementary Fig. 8) that evolves towards the formation of a plateau between Ge and Si peaks for a porosity of 70%. This phenomenon suggests a progressive accommodation of the lattice strain between both materials. Indeed, owing to the high porosity, the porous silicon pillars exhibit low Young's modulus allowing easy deformation to accommodate the lattice mismatch with the Ge microcrystals. Furthermore, the amount of diffused Ge, that may occur, into the PSiP is expected to increase with increasing the pillars' porosity and consequent decrease of the materials density. This phenomenon is expected to reduce the overall amount of pure Ge in the microcrystal and is likely to be the origin of the slight decrease of the Ge diffraction peaks intensity. Additionally, the Ge/Si intermixing mediated by porous Si reorganization during epitaxy can lead to the formation of $Si_{1-x}Ge_x$ alloy with graded composition that further contributes to reducing the lattice mismatch between the PSiP and the Ge microcrystal. Nonetheless, the well-known growth of graded layer is not enough to annihilate the lattice strain within only a few micrometers. In the present case, both phenomena, the porous structure deformation, and the interdiffusion of Si and Ge mediated by the porous structure, coexist giving rise to the observed elastic strain accommodation. While lower pillars porosities are expected to improve the Ge microcrystal's structural properties, full strain accommodation arises only when the porosity reaches 70% (Supplementary Fig. 8) suggesting a threshold porosity that may vary depending on the epitaxial material. This highlight that the porosity constitutes a key parameter to accommodate the lattice mismatch strain. Moreover, as can be seen in literature, simultaneous reorganization of both Ge and porous silicon can lead to decrease the TDD[43]. High porosity combined with our low growth rate may favored this reorganization and thus the SiGe synthesis leading to the strain accommodation.

It's worth mentioning that based on the theory of compliance[10,51] and discarding the Ge diffusion and consequent alloying effects, a simple analytical estimation of the areal strain energy associated with an isolated screw dislocation in Ge epitaxial layer on porous Si as a function of the porosity also predict an onset for full compliance around Si porosity of 68% (Supplementary Fig. 9).

## Strain relaxation of the heterostructure

To clarify the compliant properties of such substrate, strain mapping using the STEM Moiré GPA method[52] was performed. Figure 4, describes the obtained results for Ge/SiP and Ge/PSiP(70%) respectively. Using the center of the Ge microcrystal as the arbitrary zero deformation, the strain and rotation maps demonstrate that the relative deformation field is globally very low in the whole Ge region for both samples. As the lattice mismatch between Ge and Si is around 4.18%, significant strain relaxation is expected from the side of the Ge pillar if the Ge material is lattice matched to the Si underneath. The very low deformation field suggest that no strain relaxation is observed from the side of structure. Therefore, the Ge regions in both the reference and the Ge/PSiP samples are nearly fully relaxed. For the reference sample, the strain between Ge and Si pillars from the atomic misfit is released at the interface between materials with a well define misfit dislocations network and threading dislocations in the Ge microcrystal (Supplementary Figs. 10 and 11). For the Ge/PSiP sample, the Ge grows from $Si_{1-x}Ge_x$ template is able to reduce the relative atomic misfit and potentially allow the Ge mechanical constraints to be accommodated with a local elastic deformation.

The germanium grown on PSiP presents a pronounced strain state located at the proximity of the $Si_{1-x}Ge_x$ compound between the pores (Supplementary Figs. 12 and 13). Since the arbitrary zero deformation is defined taking the center of the Ge microcrystal as reference point, the lattice parameter of the material incorporated in the porous structure appears shrank in both in-plane and vertical directions. This would suggest that most of the formed $Si_{1-x}Ge_x$ alloy material seems to be partially or nearly fully relaxed. However, without knowing the composition at the precise location it is difficult to be conclusive on the strain state of these compounds. Variations of deformation are observed in the substrate which can be both related to the different alloy composition or a real elastic deformation. Accepting the limitation in the data interpretation, the STEM Moiré GPA strain mappings, with both XRD and the STEM EDX, results are in good agreement to confirm the strain accommodation due to the gradual $Si_{1-x}Ge_x$ formation and the deformation of the porous substrate allowed by its elasticity. This strain accommodation leads to the formation of defect-free microcrystals, which can be easily turned into a two dimensional layer suitable for device synthesis by increasing the deposited thickness[53].

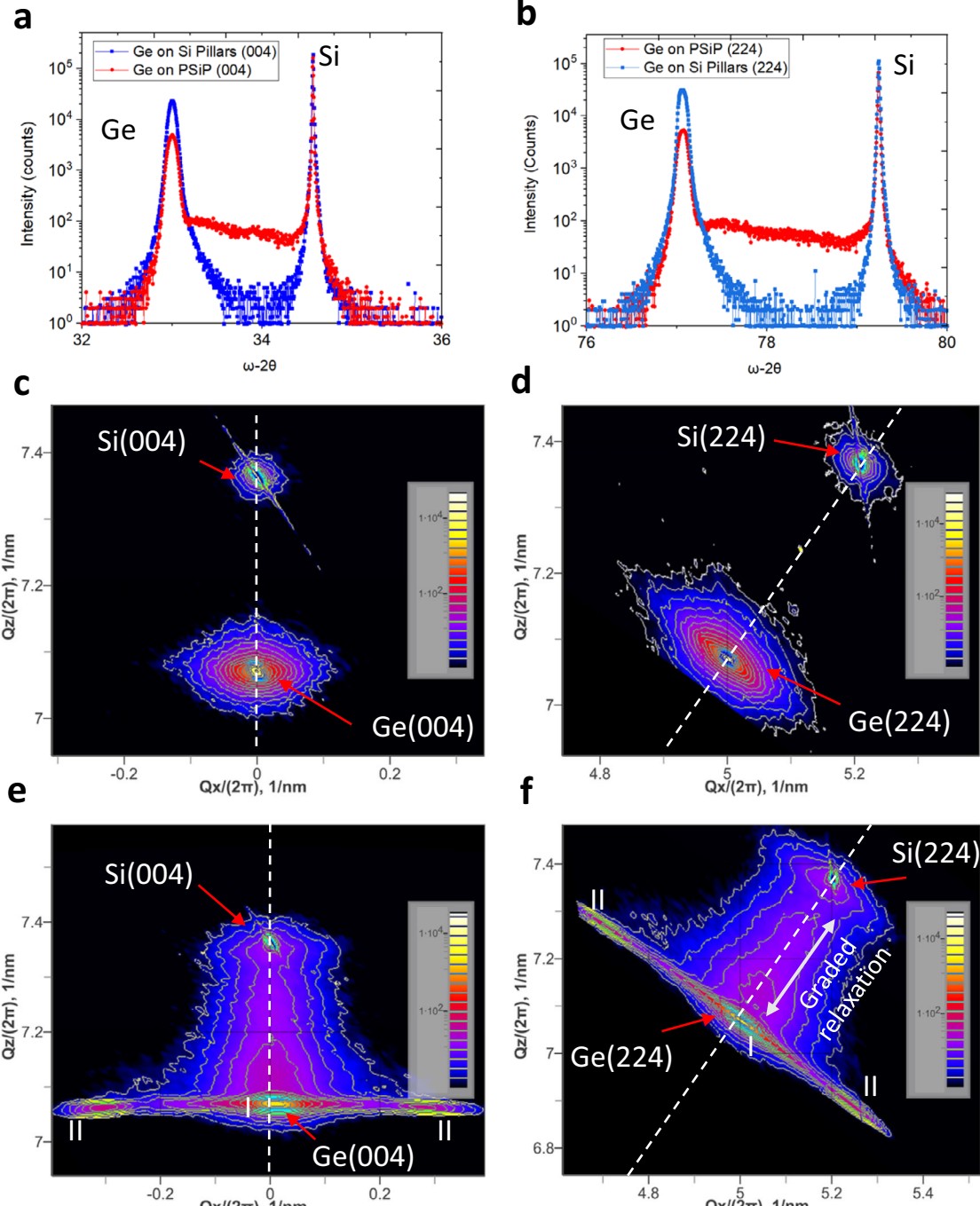

**Fig. 3 | The analysis of crystal quality and residual strain in the Ge/Si hetero-structure. a, b** Coupled scans ω−2θ of the Ge/SiP (**a**) and Ge/PSiP(70%) (**b**) heterostructures around the Si(004) and (224). **c, d** Reciprocal space mapping of the Ge/SiP heterostructures around the symmetric Si(004) and asymmetric Si(224) reflections. **e, f** Reciprocal space mapping of the Ge/PSiP heterostructures around the symmetric Si(004) and asymmetric Si(224) reflections. (I) relaxed Ge microcrystals, (II) asymmetric relaxation of the Ge microcrystals.

We report the synthesis of a fully compliant substrate for the epitaxial growth of lattice mismatch material on silicon. Combining micro-patterning of silicon substrate and mesoporous crystalline structure allow full accommodation of both lattice and thermal stress. We have shown for the Ge/Si (001) heterostructure that the porosity allows a strain accommodation, through suppressing the nucleation of dislocations via porous crystallites deformation and SiGe compound formation within the mesoporous structure. TEM observations and etch-pit counting reveal the absence of any defects in Ge microcrystals grown on PSiP while Ge on SiP present high defect density. XRD characterizations confirm the improvement of Ge crystalline quality on porous medium, and the graded relaxation between materials. EDX highlight the penetration of the Ge in the porous structure and the SiGe formation, while strain mappings confirm the strain accommodation allow by the compliant substrate.

The porous structure allows to avoid formation of any defect at the interface, which not only yields a high-quality material but also allows the synthesis of devices requiring very low thickness and would decrease the cost and time associated to the growth process of metamorphic structures.

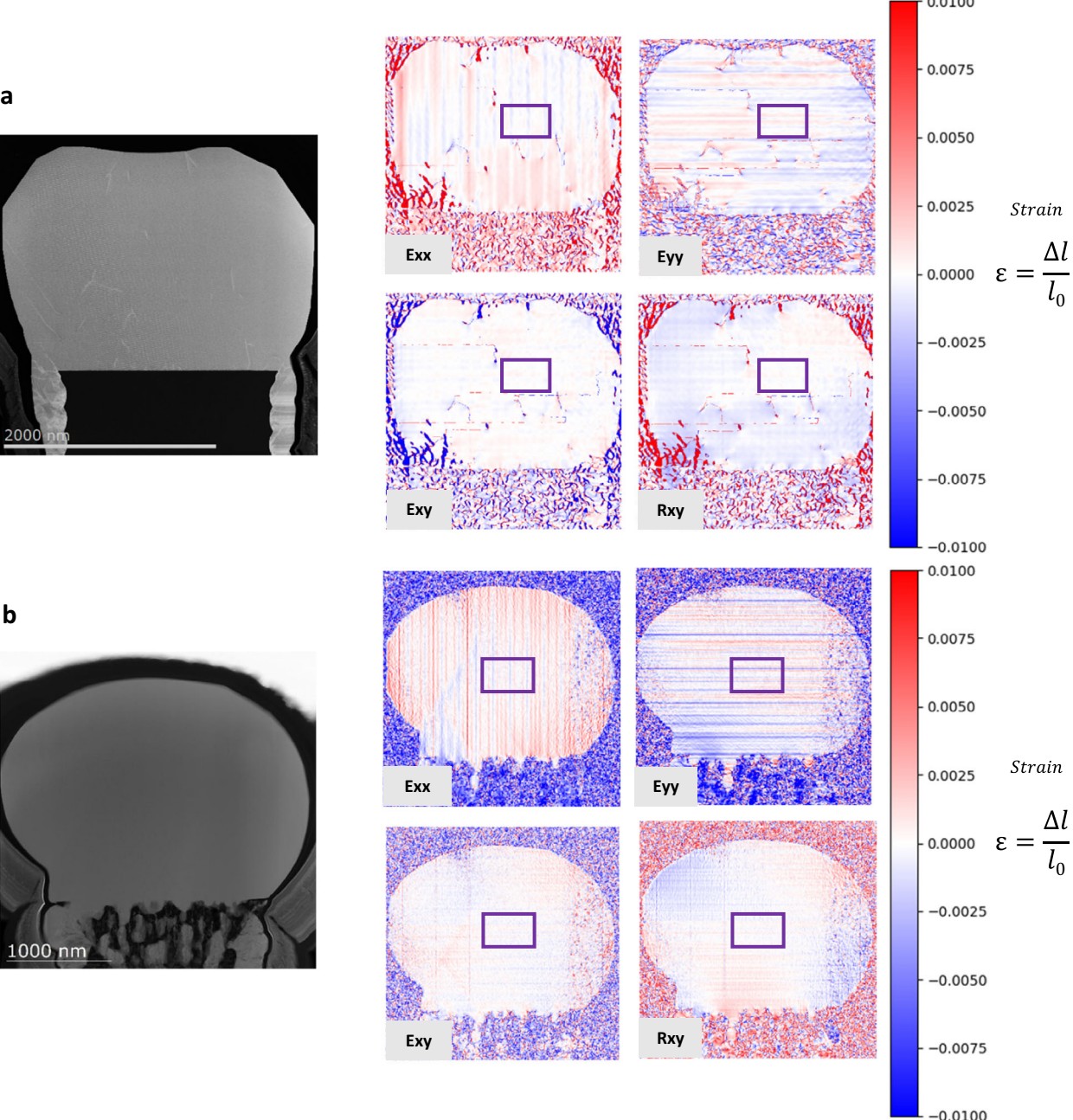

**Fig. 4 | STEM strain mapping characterization. a, b** 2D strain and rotation maps of Ge/Si pillars reference sample (**a**) and Ge/PSiP(70%) (**b**), using purple box region as the arbitrary zero deformation.

This study provides a proof of concept for the synthesis of effective compliant substrate for heteroepitaxy and the integration of lattice mismatch microcrystals on Si platform. The method used can provide similar defect-free system on large surface area and a practical template for microfabrication processes. We believe that such compliant template paves the way towards the potential synthesis of defect-free heterostructure including the direct growth of various materials such as GaAs, InP, and GaN. The pillar structure can naturally be tuned to maximize the yield of targeted devices.

## Methods

### Substrate patterning

The 400 μm thick Si(001) p-type(B) (0.01–0.02 Ohm.cm) substrates were patterned by deep reactive ion etching (DRIE) based on the Bosch process[54] with high etch rate (few micrometers per minute) and good

anisotropy. As a result, we obtain 2 μm × 2 μm square pillars, 10 μm tall, separated by 1 μm gap.

### Nanostructuration

Prior to porosification, patterned substrates were cleaned using ethanoic alcohol for 10 min follow by 5 min in isopropanol.

To perform the etching of Si pillars, a custom-made electrochemical cell of Teflon was employed. The electrochemical cell was constituted of a copper electrode as the backside wafer contact (isolated from the electrolyte), a platinum counter-electrode and the patterned substrate as the working electrode. Most of Porous Silicon Porous (PSiP) was obtained by pulsed electrochemical etching process. An anodization was carried out in an O-ring electrode with an HF:Ethanol (volume:volume) electrolyte. Cathodic current density was set up to 0 mA/cm². The substrate used was a one-side polished, B-doped,

p-type (100) Si wafer. The 400 μm-thick wafers were 5 × 5 cm² with a measured resistivity between 10 and 20 mΩ.cm. HF last process (5% diluted) pre-cleaning of Si pillars and PSIP substrates was performed to suppress native oxide ($SiO_2$) formation, then blown dry with nitrogen and introduced into the loading chamber of the CBE reactor.

### Crystal growth

Ge growth was carried out in a VG Semicon VG90H CBE reactor. A thermocouple was used for growth temperature monitoring. Ge microcrystals were grown using a solid source of Ge with a KCell temperature kept constant at 1250 °C. The base pressure in the load-locked growth chamber was below $1 \times 10^{-6}$ Torr, whereas during growth the pressure was ~$8.4 \times 10^{-6}$ Torr.

### Investigation of the porous structure, defect density, and crystal quality

The morphology and thickness of the grown microcrystals and porous structure were characterized with a LEO 1530VP scanning electron microscope (SEM) (Supplementary Fig. S2). Porosity was first determined using the ImageJ software. Accurate determination of the porosity was determined using Fourier-transform infrared spectroscopy (FTIR) (Supplementary Fig. 3). Spectra were recorded with a Hyperion 2000 FTIR microscope using a Globar source, a KBr beam splitter, and a MCT D316 detector.

The structural properties of the microcrystals were investigated by using Rigaku SmartLab system, equipped with a 2-bounces Ge (220) crystal monochromator on the incident beam. The high-resolution x-ray diffraction measurements, $\omega - 2\theta$ scans and reciprocal space maps (RSMs) were performed around the Si (004) symmetrical and (224) asymmetrical Si reflections with Cu Kα1 radiation.

The etch pit method were carried out using a mixture of 2 volumetric parts 49 wt. % HF and 1 part 0.1 M $K_2Cr_2O_7$. Etch pits were counted on the top surface by examining Top-view SEM images of different pillars.

### Fourier-transform infrared spectroscopy (FTIR)

Porous medium present different optical properties compare to the original bulk sample. Therefore, the incident infrared light on the mesoporous Si substrate will be reflected from two different interfaces: air/porous Si and porous Si/Si substrate[55]. Considering these different interfaces, the transmitted and reflected radiations will result in a Fabry–Perot interference spectrum. This interference pattern obtained with the reflective measurement is used to determine the refractive index of the porous medium ($n_{Porous}$) with the Eq. (1):

$$m\lambda_{max} = 2n_{Porous}L \tag{1}$$

where $m$ is the order of the fringe, $\lambda_{max}$ is the wavelength of the fringe maximum, $L$ the thickness of the porous structure (determined using SEM observations), and $2n_{Porous}L$ is the effective optical thickness.

Therefore, the linear equation can be used for each maximum peak:

$$m = 2n_{Porous}L\left(\frac{1}{\lambda_{max}}\right) + b \tag{2}$$

The Bruggeman Effective Medium approximation (Eq. 3) is well suited for determining the porosity of the mesoporous Si:

$$(1 - P)\frac{(n_{Si}^2 - n_{Porous}^2)}{(n_{Si}^2 + 2n_{Porous}^2)} + P\frac{(n_{Pore}^2 - n_{Porous}^2)}{(n_{Pore}^2 + 2n_{Porous}^2)} = 0 \tag{3}$$

where $P$ is the porosity and $n_{Si}$ and $n_{Pore}$ are the refractive index of Si bulk and the medium filling the pore (air), respectively.

### Samples preparation for scanning transmission electron microscopy (STEM) imaging

The samples were prepared using a Zeiss NVision 40 Focused Ion Beam (FIB). The target locations were first coated with electron-beam-induced carbon deposition to protect the surface from being sputtered off by the ion beam and followed by ion-beam-induced deposition of tungsten for further protection throughout the FIB process. The target locations were extracted from the specimen and attached to copper FIB grids using a conventional FIB milling and lift-out procedure, with the exception that the extractions were significantly taller than a typical FIB lamella. The purpose was to capture the full height of the pillars and to leave enough substrate material remaining below the pillars to maintain a thick bottom frame for structural support.

Additionally, a support frame was utilized to split the thin window into two, which is intended to reduce the impact of warping when the window thicknesses are reduced towards TEM transparency. This method also enables the lamella to possess two different thicknesses to better optimize for a wider variety of characterization techniques. The thinning was performed with the ion beam voltage at 30 kV using milling lines of progressively smaller beam currents down to 40 pA. Final cleaning steps were performed with the ion beam voltage at 10 kV and 5 kV using a glancing angle raster box with the stage tilted 8 degrees below the tilt angle of the thinning step.

### STEM imaging of Ge microcrystals and Energy-dispersive X-ray spectroscopy (EDS)

TEM/STEM observations were made on a Titan Themis 200 microscope (FEI/ Thermo Fischer Scientific) equipped with a geometric aberration corrector on the probe. The observations were made at 200 kV with a probe current of about 50 pA and a half-angle of convergence of 17 mrad. HAADF-STEM images were acquired with a camera length of 110 mm (inner/outer collection angles were respectively 69 and 200 mrad). The microscope is also equipped with the "SuperX" EDS elemental analysis system with 4 windowless EDX detectors (detection angle 0.8 steradian).

### STEM strain mapping

The STEM imaging was performed on a FEI Titan Cubed 80-300 equipped with CEOS correctors on both the probe and image forming lens systems operating at 200 keV. The STEM probe size, current, and semi-convergence angle were approximately 100 pm, 80 pA, and 19.8 mrad respectively. The STEM acquisition conditions were set to obtain Z-contrast type STEM electron micrographs with inner/outer angles of the Fischione annular dark-field (ADF) detector of 50 and 200 mrad, respectively. The strain characterization was performed using the STEM Moiré GPA method and processed using a homemade open-source python script available on a public repository[56].

## Data availability

All relevant data are available from the authors upon request.

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

## Acknowledgements

The authors would like to thank H. Pelletier, G. Bertrand, P. O. Provost, and T. Casagrande for the technical help, the Natural Sciences and Engineering Research Council of Canada (NSERC), the NSERC Award Number 497981 and the Fonds de Recherche du Quebec-Nature et Technologies (FRQNT) for financial support. The authors would also like to thank Daniel Chrastina and the EU Horizon's 2020 project microSPIRE (Grant No. 766955) for financial support. A.B. is grateful for a Discovery grant supporting this work. G.B. is grateful to NSERC for a Discovery Grant supporting this work. Some of the experimental work (sample preparation and strain characterization) was carried out at the Canadian Centre for Electron Microscopy, a national facility supported by the Canada Foundation for Innovation under the Major Science Initiative program, NSERC and McMaster University.

## Author contributions

A.H. performed the experiments, analyzed the data, and wrote the manuscript. A.B. prepared the patterned samples. A.P. carried out GPA analyses; characterization of HR-TEM was performed by A.P. and G.P. EDX was performed by G.P. The manuscript was revised by all authors. The project was planned, directed, and supervised by B.I., R.A., S.F., G.I., and A.B.

## Competing interests

The authors declare no competing interests.
