## [Peer Review File · Nature Communications]

Defect free strain relaxation of microcrystals on mesoporous patterned siliconREVIEWER COMMENTS

Reviewer #1 (Remarks to the Author):

This is an interesting paper discussing the use of treated Si micropillars as a compliant substrate. The differing results between the control SiP and PSiP are compelling and the results, in general, are well characterized and described. This reviewer has several significant comments for the authors as well as numerous smaller points made in the attached pdf. This reviewer would like further discussion and evidence about what is happening within the pillars to accommodate the lattice mismatch of the Ge tops.

Comments

1. What is the diffraction condition for the images in Figures 2c and 2d? NOTE: some diffraction conditions will make dislocation disappear.
2. In Figure 2b, the authors talk about a tilt. Could the authors explain this more? This reviewer is not certain what the authors are trying to highlight here.
3. The authors talk about the "potential scalability" of their method. Could the authors explain more about this?
4. This reviewer was uncertain what the authors meant by "exhibits facets oriented towards the center of the structure". Please explain more.
5. Do the authors have an idea as to why the Ge on PSiP have a lower Ge peak height? This reviewer would expect BOTH that the Ge peak FWHM would be reduced AND that the peak height would be greater. This is not, however the case. Is there a reduced amount of Ge in the caps for the Ge on PSiP? Do the authors have any other ideas as to why this might be?
6. For Figure 3, could the authors please make explicit which figures are for PSiP and for SiP? This is neither explicitly spoken about in the text nor the caption, though, this reviewer suspects that Figures 3e and 3f are for PSiP since they include the "graded" region.
7. Could the authors also address the effect of the small crystal size on the crystal quality, comparing their results to those in Ref. 23, for example.
8. This reviewer would suggest that the authors do SEM EDS line scans on the pillars to get a sense, albeit not very accurate, of how much Ge is present in the pillars. This may enable the authors to make some further suggestions (not conclusions) as to what is happening in the pillars. If there is really a compliant substrate effect, this reviewer would suspect that the pillar is becoming highly dislocated and/strained and some of this could be seen in the XRD data. (For example, if the authors find that, at most there is 10% Ge within the pillars, then that could be used to give an upper bound to the Ge content in the pillar and be used to say something about the strain state of the pillar which is part of the "graded relaxation region".
9. As follow up to comment 8, the authors could also perform XRD on the Si pillars, as prepared and after the initial Ge growth, 200 nm growth. This reviewer would suspect that there would not be as much relaxation in the pillars after the initial 200 nm of growth and this data could help to better explain what is happening in the pillar to accommodate the strain.
10. The authors do not discuss the asymmetric relaxation regions. Please do so. Where does it come from and what does it mean?
11. This reviewer is more accustomed to the lattice mismatch between Si and Ge as 4.01% defined as in Eq 1 of <https://www.sciencedirect.com/science/article/pii/S0920230791900069>
12. Could the authors discuss more why "the material incorporated in the porous structure is also compressed in both...directions?" This reviewer would have thought the Si pillars would be under tensions since the Ge is a larger lattice constant.

Reviewer #2 (Remarks to the Author):

Overcoming materials compatibility for epitaxy is always an interesting topic making impacts and progress in diverse field. The authors' study on 'compliant' substrate prepared by porosification of silicon pillar to grow crystalline germanium microcrystals should fit in the topic of overcoming materials compatibility. The motivation and the experimental procedures are described well. The structural characterization and analyses are reasonable. However, the authors had better include a few other details and discussion in the manuscript for epitaxy researchers to follow the authors' direction.

1) Clear description of porosification of Si pillars: According to Figs. 1 (b) and (d), the porous region is limited in the bottom regions of the pillars though the wet etching procedure would make porous regions in the whole pillars. I recommend the authors to clarify if the authors intentionally formed porous regions at the bottoms. The question brings additional questions.

If the porous regions are confined at the bottoms, 'compliant' substrate behavior would be affected by the volume of the porous region. For instance, the Si pillars can be 'compliant' substrate when the volume of the porous region is higher than certain value. The authors need to discuss the effect. If the porous regions are everywhere in pillars, the nucleation of Ge microcrystal would be affected by the multiple crystal planes exposed by the etching procedure. The authors need to discuss the effect.

2) Nucleation behavior of Ge microcrystals: According to Figs 2 (c) and (d), facet formation of the Ge crystals is different along the substrate preparation procedure. The facet formation shown in Fig. 2(c) is commonly observed. However more spherical shape shown in Figure 2(d) is not common for Ge on Si. The authors can bring fruitful discussion on the facet formation governed by nucleation on different substrates.

3) The authors insist that their progress can deliver lots of advantages for materials preparation and device manufacturing, such as high quality, long-term reliability, and combinations for completely incommensurate materials. This manuscript has proven 'high-quality' issue only. Long-term reliability would be obtained by better crystal quality, but the substrate itself looks mechanically weak. The porous substrate may generate adhesion issue and weakened robustness as shown in Figure 1(b). For other compound semiconductors, porosification hasn't been studied thoroughly like Si processing. The authors may change the tone of their conclusion.

Reviewer #3 (Remarks to the Author):

The paper describes the epitaxial growth of Ge microcrystal on deep etched porous Silicon pillars. The method provides a fully compliant substrate enabling elastic relaxation of defect free Ge microcrystals.

Although supplementary info gives partial information on the porosification process on the micropillars the process is not properly described in the main text. In particular, different porosities were obtained but the structures are not completely described.

In figure1 the porous silicon seems to be present only at the bottom of the micropillar. The description of the Si structure in between the Ge microcrystal and the bottom porous part is lacking. The Ge stands on this structure.

Finally, the authors should describe the impact of porosity on the Ge structure and dislocation density. It is expected also that porosity plays a major role on Ge diffusion in Si pores that in turn has an effect on the strain.

The paper should discuss in more detail the advantage of the micropillars with porous substrate and compare the dislocation density with the results reported in the literature

See for instance

- Ge growth on porous silicon: The effect of buffer porosity on the epilayer crystalline quality G. Calabrese et al. APL 105 (2014) 122104

- Enhanced reduction in threading dislocation density in Ge grown on porous silicon during annealing due to porous buffer reconstruction G. Calabrese et al. Phys. Stat. Sol. A 213 n°1 (2016) 96

In conclusion the paper in its actual form is incomplete. Major modifications are required.

Additional comments :

p4 : "SEM image of Bosch process deeply patterned p-type Si (001) wafers with ordered square-based 5x5 cm² arrays of Si pillars separated by 2 μ m trenches used as substrates is shown by the Fig. 1a. "

It is not 2 but 1 μ m

P6 : figure caption of figure 3 needs to be revised

"Coupled scans ω -2 θ of the Ge/SiP(a) and Ge/PSiP(70%)(b) heterostructures around the Si(004) and (224). c-f, Reciprocal space mapping of the respective heterostructures around the symmetric Si(004) and asymmetric Si(224) reflections. (I) relaxed Ge microcrystals, (II) asymmetric relaxation of the Ge microcrystals."

p8 : The very low deformation field suggest that nearly no strain relaxation is observed from the side of structure.

I guess it should be written no strain is observed

Responses to Reviewers:

We would like to thank the reviewers for their efforts in evaluating our work and for their constructive comments and suggestions. Thus, we have considered their recommendations in close details to improve our manuscript.

Please find below, the point-by-point responses to each of the reviewers' comments (highlighted in Bold) along with the responses and action taken in the manuscript. We refer to each of the comments as, for example, 'R1.2' meaning reviewer #1's second comment.

We believe that the revised manuscript, which considers the main points raised by the reviewers, has been considerably improved.

Reviewer #1 (Remarks to the Author):

This is an interesting paper discussing the use of treated Si micropillars as a compliant substrate. The differing results between the control SiP and PSiP are compelling and the results, in general, are well characterized and described. This reviewer has several significant comments for the authors as well as numerous smaller points made in the attached pdf. This reviewer would like further discussion and evidence about what is happening within the pillars to accommodate the lattice mismatch of the Ge tops.

We thank the reviewer for the appreciation of our work and for the relevant remarks and constructive suggestions.

R1.1 What is the diffraction condition for the images in Figures 2c and 2d? NOTE: some diffraction conditions will make dislocation disappear.

Our Response

Figure 2c and 2d are Brightfield TEM images with the sample observed along the $\langle 110 \rangle$ zone axis. With these conditions, we will be sensitive to the deformation field in all directions, allowing the observation of all dislocations.

Action made in the manuscript

We have modified the caption of the figure 1 as follow:

"Fig. 2 | STEM characterization of Ge microcrystals along the $\langle 110 \rangle$ zone axis. a,b, Top-view SEM image of Ge/SiP (a) and Ge/PSiP(70%) (b) after Etch-pit (EP) method. **c,d**, Low magnification BF-TEM images of the Ge/SiP reference substrate (c) and of the defect-free germanium grow on Porous silicon pillar (PSiP) (d). **e**, High resolution STEM image of the Ge/Si interface with threading dislocations and misfit network. **f**, Atomic resolution STEM image of the Ge/PSiP interface."

R1.2 In Figure 2b, the authors talk about a tilt. Could the authors explain this more? This reviewer is not certain what the authors are trying to highlight here.

Our Response

We thank the reviewer for this comment. We agree that the word "tilt" may be misleading to the message we want to convey.

Indeed, by "tilt" we wanted to highlight the bending of the porous pillars resulting from their flexibility.

Action made in the manuscript

We have changed the corresponding phrase as follow (page 5, paragraph 1):

“Bending of the towers can be observed in Fig. 2 b due to the flexibility of the porous silicon pillars.”

R1.3 The authors talk about the “potential scalability” of their method. Could the authors explain more about this?

Our Response

We believe that our method can be applied to produce large scale defect free Ge microcrystals which highlights its potential scalability for industrial production.

Action made in the manuscript

To clarify this point, we have added the following sentence (page 6, paragraph 2):

“Obtaining such defect-free microcrystals on a large-scale area, as shown by EP counting, by combining Bosch process, porosification and epitaxial growth, indicates that our method could potentially be scalable for industrial production.”

R1.4 This reviewer was uncertain what the authors meant by “exhibits facets oriented towards the center of the structure”. Please explain more.

Our Response

We thank the reviewer for bringing this out. The sentence is indeed unclear and need to be rewritten with more details

Action made in the manuscript

The sentence has been rewritten as follow (page 6, paragraph 3):

“ As we can deduct from the TEM observations (Fig. 2c) of the reference sample, the Ge top surface is partly faceted with the presence of (001) central surface, surrounded by (113) facets.”

R1.5 Do the authors have an idea as to why the Ge on PSiP have a lower Ge peak height? This reviewer would expect BOTH that the Ge peak FWHM would be reduced AND that the peak height would be greater. This is not, however the case. Is there a reduced amount of Ge in the caps for the Ge on PSiP? Do the authors have any other ideas as to why this might be?

Our Response

We thank the reviewer for rising this point. Indeed, for Ge conventionally grown on planar substrate, higher crystalline quality of a given epilayer will obviously results in a lower FWHM and higher intensity. However, in the present case we believe that a combined effects of several factors lead to the observed unexpected lower peak’s intensity from Ge/PSiP.

These factors are:

- As mentioned by the reviewer, the Ge diffusion through the PSiP pores and consequent SiGe alloy formation led to a reduced amount of “pure” Ge.

-The bending of the PSiP could also contribute to the intensity reduction by deflecting part of the diffracted beam away from the main diffraction peak

Accordingly, in the present case the XRD peaks cannot account alone for accurate quantitative analysis of the Ge material quality. For this reason, we employed also Etch-Pits counting and HR-TEM.

R1.6 For Figure 3, could the authors please make explicit which figures are for PSiP and for SiP? This is neither explicitly spoken about in the text nor the caption, though, this reviewer suspects that Figures 3e and 3f are for PSiP since they include the “graded” region.

Our Response

We thank the reviewer for this remark

Action made in the manuscript

The figures have now been explicitly identified and the caption has been modified as follow:

“Fig. 3 | The analysis of crystal quality and residual strain in the Ge/Si heterostructure. a,b, Coupled scans ω - 2θ of the Ge/SiP(a) and Ge/PSiP(70%)(b) heterostructures around the Si(004) and (224). c-d, Reciprocal space mapping of the Ge/SiP heterostructures around the symmetric Si(004) and asymmetric Si(224) reflections. e-f, Reciprocal space mapping of the Ge/PSiP heterostructures around the symmetric Si(004) and asymmetric Si(224) reflections. (I) relaxed Ge microcrystals, (II) asymmetric relaxation of the Ge microcrystals.”

R1.7 Could the authors also address the effect of the small crystal size on the crystal quality, comparing their results to those in Ref. 23, for example.

Our Response

As described in reference 23, the crystal quality is found to be influenced by the pillars/mesa top surface area. Indeed, the material quality has been shown to be improved when the area of the pillar/mesa decreases. However, the pillars’ dimension used in this reference are very large (between 400 and 25 μm) compared to our patterned structure. Moreover, the material deposited, such as InGaAs and SiGe, are not highly mismatched to the substrate (0.4% and 0.6 % respectively) compared to our study (>4%).

A study involving different size of pillars comparable to ours has been published by *Rovaris, et al* ([doi:10.1103/PhysRevMaterials.1.073602](https://doi.org/10.1103/PhysRevMaterials.1.073602)). They observed, using SiGe graded layer, that the TDD decreases with decreasing the pillar width, down to a complete elimination which is a comparable trend as ref. 23.

As reported by the paper “*Falub, C. V. et al. Scaling hetero-epitaxy from layers to three-dimensional crystals. Science 335, 1330–1334 (2012)*” involving the growth of Ge on the same pillar width presented in our study, the TDD remains close to the one highlighted in our reference sample. Usually, in order to decrease the TDD and so increase crystal quality with such a pattern, specific growth conditions are required to ensure particular faceting of the microcrystals allowing dislocations expulsion towards the sidewalls.

In the present work, we highlight the ability to obtain defect-free heterostructure based on the pillar's porosity variation. Our aim is to demonstrate that strain management (using porosification) can avoid defect nucleation in highly mismatched materials

R1.8 This reviewer would suggest that the authors do SEM EDS line scans on the pillars to get a sense, albeit not very accurate, of how much Ge is present in the pillars. This may enable the authors to make some further suggestions (not conclusions) as to what is happening in the pillars. If there is really a compliant substrate effect, this reviewer would suspect that the pillar is becoming highly dislocated and/strained and some of this could be seen in the XRD data. (For example, if the authors find that, at most there is 10% Ge within the pillars, then that could be used to give an upper bound to the Ge content in the pillar and be used to say something about the strain state of the pillar which is part of the “graded relaxation region.

Our Response

We thank the reviewer for the pertinent suggestion. Indeed, as shown by the supplementary figure 6, the EDX characterizations highlighted a graded infiltration of Ge in the porous structure all over the porous pillars. Such graded compound exposes a progressive strain state resulting in the observed plateau between Si and Ge peaks in XRD. Moreover, in the RSM (Figures 3d and 3f), the Si peak broadening increases with the porosity. We agree with the reviewer that this would indicate that dislocations may arise in the PSiP because of the strain transfer from the epitaxial material. Meanwhile, further structural investigations by TEM will be needed to conclude about the formation of dislocation.

R1.9 As follow up to comment 8, the authors could also perform XRD on the Si pillars, as prepared and after the initial Ge growth, 200 nm growth. This reviewer would suspect that there would not be as much relaxation in the pillars after the initial 200 nm of growth and this data could help to better explain what is happening in the pillar to accommodate the strain.

Our Response

We agree with the reviewer that the study of the strain relaxation for different Ge thickness would be very interesting for deeper understanding of the strain accommodation and are aware about the importance of the proposed work. Indeed, since the buffer layer of 200 nm has been deposited at low growth temperature, no reorganization of the porous Si pillars is expected at this stage. A complete and detailed study of the initial growth nucleation and Ge diffusion kinetics in porous Si pillars as a function of the thermal budget will be considered in our future work. Indeed, since the XRD cannot account alone for a quantitative analysis as the Ge deposited in the sidewall as well as in between the pillars will complicate the analysis. More accurate characterization such as TEM and EDX will also be considered.

R1.10 The authors do not discuss the asymmetric relaxation regions. Please do so. Where does it come from and what does it mean?

Our Response

Thank you for the comment.

We agree that this specific aspect could be not clear.

Following the literature (Falub, C. V. et al. Scaling hetero-epitaxy from layers to three-dimensional crystals. *Science* 335, 1330–1334 (2012)), last Ge towers of each pillar blocks are not subject to the

same vertical growth due to missing neighbours. As a result, we obtain Ge microcrystals partially strained.

R1.11 This reviewer is more accustomed to the lattice mismatch between Si and Ge as 4.01% defined as in Eq 1 of <https://www.sciencedirect.com/science/article/pii/S0920230791900069>

Our Response

We understand the reviewer concerns as discrepancies may arise between the reported values for the lattice mismatch between Si and Ge in the literature. However, most of the available literatures and more precisely the most recent ones report the lattice mismatch between Ge and Si as 4.2%. See for example:

_ Nathan Newman, Mahmoud Vahidi, in Handbook of Crystal Growth: Thin Films and Epitaxy (Second Edition), 2015, 20.3.3.1 Brief Case Study of Nucleation Controlled Growth: Ge/Si (100)

_ Falub, C. V. *et al.* Scaling hetero-epitaxy from layers to three-dimensional crystals. *Science* **335**, 1330–1334 (2012).

R1.12 Could the authors discuss more why “the material incorporated in the porous structure is also compressed in both...directions?” This reviewer would have thought the Si pillars would be under tension since the Ge is a larger lattice constant.

Our Response

We thank the reviewer for this remark.

Indeed, considering the relaxed Ge microcrystal as reference (purple box in Fig. 4), we employed the sentence “compressed in both directions” to qualify the observed shrinkage of the pillar’s alloy material’s lattice parameter in both in-plane and vertical direction. We agree with the reviewer that this can be confusing. Hence, the text has been modified as described below.

Action made in the manuscript

In order to clarify this point, we modified the text as follow (page 10, paragraph 1):

“Since the arbitrary zero deformation is defined taking the center of the Ge microcrystal as reference point, the lattice parameter of the material incorporated in the porous structure appears shrank in both in-plane and vertical directions. This would suggest that most of the formed $\text{Si}_{1-x}\text{Ge}_x$ alloy material seems to be partially or nearly fully relaxed”

Reviewer #2 (Remarks to the Author):

Overcoming materials compatibility for epitaxy is always an interesting topic making impacts and progress in diverse field. The authors' study on 'compliant' substrate prepared by porosification of silicon pillar to grow crystalline germanium microcrystals should fit in the topic of over coming materials compatibility. The motivation and the experimental procedures are described well. The structural characterization and analyses are reasonable. However, the authors had better include a few other details and discussion in the manuscript for epitaxy researchers to follow the authors' direction.

We are grateful to the reviewer for the appreciation of our work and want to thank him for the positive assessment and constructive suggestions

R2.1-a Clear description of porosification of Si pillars: According to Figs. 1 (b) and (d), the porous region is limited in the bottom regions of the pillars though the wet etching procedure would make porous regions in the whole pillars. I recommend the authors to clarify if the authors intentionally formed porous regions at the bottoms. The question brings additional questions. If the porous regions are confined at the bottoms, 'compliant' substrate behavior would be affected by the volume of the porous region. For instance, the Si pillars can be 'compliant' substrate when the volume of the porous region is higher than certain value. The authors need to discuss the effect.

Our Response

We understand that the SEM image in the figure 1, showing only the bottom part of the porosified pillar, can be confusing. Indeed, the upper part of the pillars cannot be observed due to a screening effect coming from the Ge deposition on the lateral surface of the pillars. Meanwhile, the Si pillars have been entirely porosified as can be observed on TEM images (Supplementary Material: Fig. S5(c) and S5(d)) testifying that the porous region extends all over the pillar's height.

Fig. S5. TEM characterization of the heterostructures. Low magnification BF-TEM images of the Ge/SiP reference sample (a), high resolution TEM image of the Ge/SiP interface with misfit dislocation network (b). Low magnification BF-TEM image of several Porous silicon Pillars (PSiP) substrates with a defect-free Ge crystals (c), the porous structure (d). Selected area electron diffraction (SAED) pattern of the relaxed and monocrystalline Ge crystal on the porous structure (e), and the SAED pattern of the porous silicon (f). TEM observations were performed on several pillars for each Ge grown on SiP and PSiP substrates. The defects appeared in the reference sample with the dislocation misfit network, while Ge microcrystals grown on PSiP substrate remain defect-free. SAED patterns confirm the

monocrystalline and relaxed Ge microcrystal obtained on PSiP. Moreover, the deformation of the dots in the SAED pattern related to the substrate shows the lattice accommodation of the porous template and SiGe compound.

Action made in the manuscript

In order to clarify the porosification of Si pillars, we have modified the caption of Figs. 1 as follow:

“Fig. 1 | Compliant substrate realized using patterned Si(001) porous substrate. a, Cross section SEM image of 10 μm tall deeply patterned silicon wafer using Bosch process. **b,** Cross section SEM image of the bottom part of anodized silicon pillars, **the upper part of the porous Si pillars cannot be observed by SEM due to the screening effect coming from the Ge deposition on the lateral surface and the cleavage, however the TEM analysis reported in the Supplementary Material shows the reorganized porous structure over the entire Si pillar. c-d,** 2 μm tall Self-limited Ge microcrystals grown at 500°C by Chemical Beam Epitaxy using Ge **solid source** on SiP (c) and on PSiP(70%) (d).”

R2.1-b If the porous regions are everywhere in pillars, the nucleation of Ge microcrystal would be affected by the multiple crystal planes exposed by the etching procedure. The authors need to discuss the effect

Our Response

We agree with the reviewer on this point. Indeed, the electrochemical etching of the SiP considerably affects the nucleation of the Ge microcrystals. The presence of Si nanocrystals acts as nucleation sites for Ge nucleation resulting in different faceting of the Ge microcrystals compared to those on PSiP. Further details are provided as a response to the next comment

R2.2 Nucleation behavior of Ge microcrystals: According to Figs 2 (c) and (d), facet formation of the Ge crystals is different along the substrate preparation procedure. The facet formation shown in Fig. 2(c) is commonly observed. However more spherical shape shown in Figure 2(d) is not common for Ge on Si. The authors can bring fruitful discussion on the facet formation governed by nucleation on different substrates.

Our Response

We thank the reviewer for highlighting this point.

Indeed, the facet formation exists in all the investigated structures regardless the substrate preparation procedure as shown by the Top view SEM images (Supplementary Fig 4). In the meanwhile, the pillars porosification is found to induce faceting morphological changes. The difference in morphology can be attributed to the lack of a unique/dominant plan due to the porous template. The porous silicon pillars exhibit crystallites, each of them constitutes a nucleation site for Ge crystal resulting in a reduced area of $\langle 001 \rangle$ facet in favour of increasing that in the other crystallographic orientations.

The appearance of a more rounded crystal shape is actually expected as crystallites of group IV elements approach thermodynamical equilibrium. Most of the work carried out so far on Ge microcrystals grown on patterned substrate has been performed using low energy plasma-enhanced CVD, which due to its high deposition rates operates in out-of-equilibrium condition. However, upon prolonged annealing steps or using thermal CVD more rounded crystal shapes have indeed been

observed (see ACS Appl. Mater. Interfaces 2015, 7, 19219–19225 and ACS Appl. Mater. Interfaces 2016, 8, 26374–26380). CBE, used in this work, also features much slower growth rates resulting in more rounded crystal shape.

Action made in the manuscript

We have added our answer in the manuscript (page 6, paragraph 1), as well as this new figure in the supplementary information:

“A full elastic strain accommodation should be accompanied by the absence of TDs. To validate this, etch-pit (EP) counting has been performed on both Ge epitaxial material deposited on SiP and PSiP. For this purpose, samples were immersed in a solution of two volumetric parts 49 wt% HF and 1 part 0.1 M K₂Cr₂O₇, where mixed and screw dislocations in the Ge material get selectively etched allowing their quantification by using plan view SEM observations. The average defect density has been extracted from different Ge/SiP and Ge/PSiP top-view SEM images. As shown by the Fig. 2a, relatively high TDD around $5 \cdot 10^8 \text{ cm}^{-2}$ is found to reach the surface for Ge grown on SiP. Meanwhile, for Ge grown on PSiP, the surface appears completely free of pits implying that no TD dislocation reaches the surface (Fig. 2b). Bending of the towers can be observed in Fig. 2b due to the flexibility of the porous silicon pillars. To assess the impact of intermediate porosity on the TDD, Etch-pit counting on Ge microcrystals grown on 50% PSiP (Supplementary Fig. 4) revealed that the TDD decreases down to $2.5 \cdot 10^8 \text{ cm}^{-2}$. The decrease of the threading dislocation density confirms that the crystalline quality of the Ge microcrystal's gets progressively improved with increasing the PSiP porosity.

The facet formation has been observed in all the investigated structures regardless the substrate preparation procedure as shown by the Top view SEM images (Supplementary Fig 4). In the meanwhile, the pillars porosification is found to induce faceting morphological changes. The difference in morphology can be attributed to the lack of a unique/dominant plan due to the porous template. The porous silicon pillars exhibit crystallites, each of them constitutes a nucleation site for Ge crystal resulting in a reduced area of <001>facet in favour of increasing that in the other crystallographic orientations. In some cases (see e.g Fig 2c) the microcrystals feature a rounded morphology. This is indeed expected in group IV crystallites at thermodynamic equilibrium⁴⁹.”

Fig. S4. Etch-Pit method of Ge microcrystal on various substrate. TOP view SEM images of Ge/SiP(a), Ge/PSiP(50%)(b),Ge/PSiP(70%)(c), after Etch-Pit method.

R2.3 The authors insist that their progress can deliver lots of advantages for materials preparation and device manufacturing, such as high quality, long-term reliability, and combinations for completely incommensurate materials. This manuscript has proven 'high-quality' issue only. Long-term reliability would be obtained by better crystal quality, but the substrate itself looks

mechanically weak. The porous substrate may generate adhesion issue and weakened robustness as shown in Figure 1(b). For other compound semiconductors, porosification hasn't been studied thoroughly like Si processing. The authors may change the tone of their conclusion.

Thank you for this comment.

Action made in the manuscript

Following the reviewer recommendation, we have changed the conclusions as follow:

“The porous structure allows to avoid formation of any defect at the interface, which not only yields a high-quality material but also **allows the synthesis of devices requiring very low thickness and would decrease the cost and time associated to the growth process of metamorphic structures.**

This study provides a proof of concept for the synthesis of effective compliant substrate for heteroepitaxy and the integration of lattice mismatch microcrystals on Si platform. The method used can provide similar defect-free system on large surface area and a practical template for microfabrication processes. We believe that such compliant template **paves the way towards the potential synthesis of defect-free heterostructure including the direct growth of various materials such as GaAs, InP and GaN.** The pillar structure can naturally be tuned to maximize the yield of **targeted devices.**”

Reviewer #3 (Remarks to the Author):

The paper describes the epitaxial growth of Ge microcrystal on deep etched porous Silicon pillars. The method provides a fully compliant substrate enabling elastic relaxation of defect free Ge microcrystals.

We thank the Reviewer for the positive evaluation of our work. On the following we provide a detailed response to the comments and an updated version of the manuscript including the reviewer’s recommendations.

R3.1 Although supplementary info gives partial information on the porosification process on the micropillars the process is not properly described in the main text. In particular, different porosities were obtained but the structures are not completely described.

Action made in the manuscript

We have added new details about the porous structure in the main text (page 4, paragraph 1) as follow:

“SEM image of Bosch process deeply patterned p-type Si (001) wafers (**10-20 mOhm.cm**) with ordered square-based $5 \times 5 \text{ cm}^2$ arrays of Si pillars separated by $1 \mu\text{m}$ trenches used as substrates is shown by the Fig. 1a.

The obtained Si pillars were anodized (Fig. 1b) in O-ring electrochemical cell with an electrolyte composed of 1:3 volume ratio of HF (49%) and anhydrous ethanol and a $50\text{mA}/\text{cm}^2$ current density to form a **$2\mu\text{m}$ thick mesoporous Si pillars with $70 \pm 5\%$ porosity. The electrochemical etching of silicon pillars leads to the formation of a dendritic morphology perpendicular to each free surface exposed by the patterned substrate (Supplementary Fig. S1).**”

We have also added more detail about the porosification process in the method (page 11, paragraph 2) as follow:

“Prior to porosification, patterned substrates were cleaned using ethanoic alcohol for 10 minutes follow by 5 minutes in isopropanol. To perform the etching of Si pillars, a custom-made electrochemical cell of Teflon was employed. The electrochemical cell was constituted of a copper electrode as the backside wafer contact (isolated from the electrolyte), a platinum counter-electrode and the patterned substrate as the working electrode. Most of Porous Silicon Porous (PSiP) was obtained by pulsed electrochemical etching process. An anodization was carried out in an O-ring electrode with an HF:Ethanol (volume:volume) electrolyte. Cathodic current density was set up to 0 mA/cm². The substrate used was a one-side polished, B-doped, p-type (100) Si wafer. The 400 μm thick wafers were 5x5 cm² with a measured resistivity between 10 and 20 mΩ.cm. HF last process (5% diluted) pre-cleaning of Si pillars and PSiP substrates was performed to suppress native oxide (SiO₂) formation, then blown dry with nitrogen and introduced into the loading chamber of the CBE reactor.”

Moreover, we have indicated the time of each porosification in Supplementary information (table S1):

Table S1. *Porosification parameters.*

Current density (mA/Cm ²)	Electrolyte HF:Eth (Volume :Volume)	Mode	Porosity	Time (s)
50	1:1	Pulse 1s/1s	40-45 %	60
120	1:1	Pulse 1s/1s	50-55%	35
50	1:3	direct	70-75%	150

R3.2 In figure1 the porous silicon seems to be present only at the bottom of the micropillar. The description of the Si structure in between the Ge microcrystal and the bottom porous part is lacking. The Ge stands on this structure.

Our Response

We thank the reviewer for this comment. Indeed, the Si pillars have been entirely porosified as can be observed on cross section TEM images (Fig. S5(c), S6(a) and S7(a)). However, after Ge growth, the upper part of the porous Si pillars cannot be observed by SEM due to the screening effect coming from the Ge deposition on the lateral surface. For this reason, the description of the Si structure in between the Ge microcrystal and the bottom porous has been made later in the manuscript (starting from the last paragraph of the page 6) based on cross section TEM, XRD and EDX analysis (Supplementary Fig. S6 and S7). Accordingly, during the epitaxial growth, Ge atoms are found to diffuse into the PSiP forming SiGe alloy with graded composition. The alloying effect is mediated by the porous Si reorganization and Si/Ge intermixing. Further explanation of the reorganized pillars alloy materials' properties has now been included in the corresponding section.

Action made in the manuscript

- We have modified the caption of Fig. 1 as follow, so that the readers can be guided to the appropriate description of the reorganized PSiP:

“Fig. 1 | Compliant substrate realized using patterned Si(001) porous substrate. a, Cross section SEM image of 10 μm tall deeply patterned silicon wafer using Bosch process. **b,** Cross section SEM image of the bottom part of anodized silicon pillars, the upper part of the porous Si pillars cannot be observed by SEM due to the screening effect coming from the Ge deposition on the lateral surface and the cleavage, however the TEM analysis reported in the Supplementary Material shows the reorganized porous structure over the entire Si pillar. **c-d,** 2 μm tall Self-limited Ge microcrystals grown at 500°C by Chemical Beam Epitaxy using Ge solid source on SiP (c) and on PSiP(70%) (d).”

- The following additional details on the reorganized pillars’ alloy materials’ properties have been included (page 8, paragraph 2)

“As can be observed on the coupled scans (Fig. 3a,b), the full width at half maximum (FWHM) of the Ge peak grown on the porous structure is narrower than the one obtained with the reference on SiP, which confirm the improvement of the crystalline quality using such compliant substrate. Additionally, in case of Ge growth on PSiP an asymmetric broadening occurs for both Ge and Si diffraction peaks (Supplementary Fig. 8) that evolves towards the formation of a plateau between Ge and Si peaks for a porosity of 70%. This phenomenon suggests a progressive accommodation of the lattice strain between both materials. Indeed, owing to the high porosity, the porous silicon pillars exhibit low Young’s modulus allowing easy deformation to accommodate the lattice mismatch with the Ge microcrystals. Furthermore, the amount of diffused Ge, that may occur, into the PSiP is expected to increase with increasing the pillars’ porosity and consequent decrease of the materials density. This phenomenon is expected to reduce the overall amount of pure Ge in the microcrystal and is likely to be the origin of the slight decrease of the Ge diffraction peaks intensity. Additionally, the Ge/Si intermixing mediated by porous Si reorganization during epitaxy can lead to the formation of $\text{Si}_{1-x}\text{Ge}_x$ alloy with graded composition that further contributes to reducing the lattice mismatch between the PSiP and the Ge microcrystal. Nonetheless, the well-known growth of graded layer is not enough to annihilate the lattice strain within only a few micrometers. In the present case, both phenomena, the porous structure deformation, and the interdiffusion of Si and Ge mediated by the porous structure, coexist giving rise to the observed elastic strain accommodation. While lower pillars porosities are expected to improve the Ge microcrystal’s structural properties, full strain accommodation arises only when the porosity reaches 70% (Supplementary Fig. 8) suggesting a threshold porosity that may vary depending on the epitaxial material. This highlight that the porosity constitutes a key parameter to accommodate the lattice mismatch strain. Moreover, as can be seen in literature, simultaneous reorganisation of both Ge and porous silicon can lead to decrease the TDD⁴³. High porosity combined with our low growth rate may favoured this reorganization and thus the SiGe synthesis leading to the strain accommodation.”

R3.3.a Finally, the authors should describe the impact of porosity on the Ge structure and dislocation density.

Our Response

We thank the reviewer for this comment.

Indeed, following the recommendation, we have performed new etch-pit counting on Ge/PSiP(50%). The results show a reduced TDD suggesting an improved crystalline property of the Ge microcrystals for intermediate porosity but still below the threshold porosity allowing full strain accommodation. As we can observe, the defect density decreases with increasing the pillars porosity which attest the capacity of this compliant substrate to accommodate the strain.

The obtained values of TDD are presented in the following table:

Porosity (%)	TDD/cm ²
0	5E+08
50	2.5E+08
70	0

Moreover, we employed an analytical model to help understanding the impact of the porosity variation on the TDD and to provisionally assess the porosity required to avoid defect nucleation based on the theory of compliance. Accordingly, discarding the Ge diffusion and consequent alloying effects, a simple analytical estimation of the areal strain energy associated with an isolated screw dislocation in Ge epitaxial layer on porous Si as a function of the porosity also predict an onset for full compliance around Si porosity of 68%.

Action made in the manuscript

- We have added this new figure in the supplementary information:

Fig. S4. Etch-Pit method of Ge microcrystal on various substrate. TOP view SEM images of Ge/SiP(a), Ge/PSiP(50%)(b), Ge/PSiP(70%)(c), after Etch-Pit method.

The associated following paragraph has also been added to the manuscript (page 5, paragraph 1)

“A full elastic strain accommodation should be accompanied by the absence of TDs. To validate this, etch-pit (EP) counting has been performed on both Ge epitaxial material deposited on SiP and PSiP. For this purpose, samples were immersed in a solution of two volumetric parts 49 wt% HF and 1 part 0.1 M K₂Cr₂O₇, where mixed and screw dislocations in the Ge material get selectively etched allowing their quantification by using plan view SEM observations. The average defect density has been extracted from different Ge/SiP and Ge/PSiP top-view SEM images. As shown by the Fig. 2a, relatively high TDD around 5.10^8 cm⁻² is found to reach the surface for Ge grown on SiP. Meanwhile, for Ge grown on PSiP, the surface appears completely free of pits implying that no TD dislocation reaches the surface (Fig. 2b). Bending of the towers can be observed in Fig. 2b due to the flexibility of the porous silicon pillars.

To assess the impact of intermediate porosity on the TDD, Etch-pit counting on Ge microcrystals grown on 50% PSiP (Supplementary Fig. 4) revealed that the TDD decreases down to $2.5 \cdot 10^8 \text{ cm}^{-2}$. The decrease of the threading dislocation density confirms that the crystalline quality of the Ge microcrystal's gets progressively improved with increasing the PSiP porosity."

- We have also added the following sentences to describe the modelling results in the manuscript (page 8, paragraph 3):

"It's worth mentioning that based on the theory of compliance^{10,51} and discarding the Ge diffusion and consequent alloying effects, a simple analytical estimation of the areal strain energy associated with an isolated screw dislocation in Ge epitaxial layer on porous Si as a function of the porosity also predict an onset for full compliance around Si porosity of 68%."

- The details for the employed model, listed below, are included in the supplementary material:

"In order to assess the effect of porosity on TDD, we employed an analytical model allowing to determine the epilayer and dislocation energy evolution depending on the porosity. With such model we can predict, from the theory of compliance^{10,51}, what would be the ideal porosity required to avoid defect nucleation.

The areal strain energy associated with an isolated screw dislocation is given by¹⁰:

$$E_D = \frac{Gb^2}{8\pi\sqrt{2}a(x)} \ln\left(\frac{h}{b}\right)$$

We First supposed an epitaxial layer grown on a compliant substrate with lattice mismatch strain $f = (a_s - a_e)/a_e$, where f is the lattice mismatch strain existing in a coherently strain epilayer, a_s and a_e are respectively the lattice parameter of the substrate and epitaxial layer. In our case we did not consider the curvature at the interface between materials.

Without such curvature, the compliant substrate and the epilayer (microcrystal) are oppositely strained: $\epsilon_{\text{epi}} - \epsilon_{\text{sub}} = f$

Where ϵ_{epi} and ϵ_{sub} represent respectively the in-plane strains in the microcrystal and the substrate. Force balance requires that:

$\sigma_{\text{epi}}h_{\text{epi}} + \sigma_{\text{sub}}h_{\text{sub}} = 0$, where h_{epi} and h_{sub} are the thickness of the microcrystal and substrate respectively, and σ_{epi} and σ_{sub} are the in-plane stresses.

The shear modulus of the porous silicon G_p and its Poisson's ratio ν_p can be described by the following empirical rules³⁸:

$$G_p = G(1 - P)^4$$

$$\nu_p = \nu(1 - P)$$

Where G and ν are respectively the shear modulus and Poisson's ration of silicon and P the porosity

Considering the elastic constant K , stress can be related to strain as follows

$$\sigma_{\text{epi}} = K_{\text{epi}} \epsilon_{\text{epi}}$$

$$\sigma_{sub} = K_{sub} \epsilon_{sub}$$

with

$$K_{epi} = 2G_{epi} \frac{(1+\nu_{epi})}{(1-\nu_{epi})}, \text{ and } K_{sub} = 2G_p \frac{(1+\nu_p)}{(1-\nu_p)}$$

The strain accumulated in the epilayer (Ge microcrystal) will be equal to:

$$\epsilon_{epi} = f \frac{K_{sub} h_{sub}}{K_{sub} h_{sub} + K_{epi} h_{epi}}$$

And the elastic energy stored per unit area of the interface can be defined as:

$$E_{int} = \frac{K_{epi} h_{epi} K_{sub} h_{sub}}{K_{epi} h_{epi} + K_{sub} h_{sub}} f^2$$

Fig. S9. Analytical model based on theory of compliance. Evolution of the Ge microcrystal and dislocation strain energy depending on the porosity.

As we can observe in Fig. S9, the porosity required to avoid defect nucleation (the strain energy accumulated in the epilayer being inferior to the energy of a dislocation) is around 68%.”

R3.3.b It is expected also that porosity plays a major role on Ge diffusion in Si pores that in turn has an effect on the strain.

Our response

We thank the reviewer for having raised this point. Indeed, the increase of the pillars' porosity is associated with a decreased material's density which in turns is expected to allow more Ge diffusion. This effect can be perceptible through the XRD analysis (supplementary figure S8). Accordingly, we have added the corresponding explanation as detailed below.

Action made in the manuscript

We have added the following details in the manuscript (page 8, paragraph 2):

“As can be observed on the coupled scans (Fig. 3a,b), the full width at half maximum (FWHM) of the Ge peak grown on the porous structure is narrower than the one obtained with the reference on SiP, which confirm the improvement of the crystalline quality using such compliant substrate. Additionally, in case of Ge growth on PSiP an asymmetric broadening occurs for both Ge and Si diffraction peaks (Supplementary Fig. 8) that evolves towards the formation of a plateau between Ge and Si peaks for a porosity of 70%. This phenomenon suggests a progressive accommodation of the lattice strain between both materials. Indeed, owing to the high porosity, the porous silicon pillars exhibit low Young's modulus allowing easy deformation to accommodate the lattice mismatch with the Ge microcrystals. Furthermore, the amount of diffused Ge, that may occur, into the PSiP is expected to increase with increasing the pillars' porosity and consequent decrease of the materials density. This phenomenon is expected to reduce the overall amount of pure Ge in the microcrystal and is likely to be the origin of the slight decrease of the Ge diffraction peaks intensity. Additionally, the Ge/Si intermixing mediated by porous Si reorganization during epitaxy can lead to the formation of Si_{1-x}Ge_x alloy with graded composition that further contributes to reducing the lattice mismatch between the PSiP and the Ge microcrystal. Nonetheless, the well-known growth of graded layer is not enough to annihilate the lattice strain within only a few micrometers. In the present case, both phenomena, the porous structure deformation, and the interdiffusion of Si and Ge mediated by the porous structure, coexist giving rise to the observed elastic strain accommodation. While lower porosities are expected to improve the Ge microcrystal's structural properties, full strain accommodation arises only when the porosity reaches 70% (Supplementary Fig. 8) suggesting a threshold porosity that may vary depending on the epitaxial material. This highlights that the porosity constitutes a key parameter to accommodate the lattice mismatch strain.”

R3.3.c The paper should discuss in more detail the advantage of the micropillars with porous substrate and compare the dislocation density with the results reported in the litterature See for instance

- Ge growth on porous silicon: The effect of buffer porosity on the epilayer crystalline quality G. Calabrese et al. APL 105 (2014) 122104

- Enhanced reduction in threading dislocation density in Ge grown on porous silicon during annealing due to porous buffer reconstruction G. Calabrese et al. Phys. Stat. Sol. A 213 n°1 (2016)96

Our response

The proposed references are interesting and are among the first attempts highlighting the potentiality of employing porous Si to improve the Ge crystal quality. These references are now included in the references of this paper

Actions made in the manuscript

1. we have changed the introduction and added the suggested references [42] and [43] as follow (pages 2, paragraph 5):

“Porous silicon is a promising material to reach such a compliance. Indeed, its mechanical properties can be tuned depending on the porosity, leading to an elastic material with low Young’s modulus while remaining crystalline^{37,38}. Several studies have been performed on the growth of GaAs, SiGe or Ge on mesoporous silicon^{39–41}. However, these studies did not highlight the compliant properties of standard porous silicon substrate. Only a slight improvement of crystalline quality has been noticed. Ge deposition on planar porous Si substrate has already been reported, the TDD has been shown to be reduced down to $2,4 \cdot 10^7 \text{ cm}^2$ after annealing steps^{42,43}. Free standing graphene mesoporous Si membrane has already been proposed as complaint substrate for the growth of GaN with high potentiality to accommodate the strain energy during epitaxy⁴⁴. Nevertheless, the effectiveness of conventional porous silicon as a suitable compliant substrate is limited by the reorganization of the porous structure during the epitaxial process^{45–47} involving high temperature, the lattice accommodation between the substrate and the epilayer or even by the brittleness of the porous silicon membrane. None of the employed methods, such as porous template, patterned substrate or even SiGe graded layer can account alone for complete suppression of TD.”

2. we have added the following sentence in the manuscript (page 8, end of paragraph 2):

“Moreover, as can be seen in literature, simultaneous reorganisation of both Ge and porous silicon can lead to decrease the TDD⁴³. High porosity combined with our low growth rate may favoured this reorganization and thus the SiGe synthesis leading to the strain accommodation.”

In conclusion the paper in its actual form is incomplete. Major modifications are required.

Additional comments :

p4 : "SEM image of Bosch process deeply patterned p-type Si (001) wafers with ordered square-based 5x5 cm² arrays of Si pillars separated by 2 μm trenches used as substrates is shown by the Fig. 1a. "

It is not 2 but 1μm

Thank you for the additional comment, we have modified the text as following:

“SEM image of Bosch process deeply patterned p-type Si (001) wafers with ordered square-based 5x5 cm² arrays of Si pillars separated by 1 μm trenches used as substrates is shown by the Fig. 1a.”

P6 : figure caption of figure 3 needs to be revised "Coupled scans ω -2 θ of the Ge/SiP(a) and Ge/PSiP(70%)(b) heterostructures around the Si(004)and (224). c-f, Reciprocal space mapping of the respective heterostructures around the symmetric Si(004) and asymmetric Si(224) reflections. (I) relaxed Ge microcrystals, (II)asymmetric relaxation of the Ge microcrystals."

Our Response

We thank the reviewer for this remark

Action made in the manuscript

We have modified the caption of the figure 3 as follow:

“Fig. 3 | The analysis of crystal quality and residual strain in the Ge/Si heterostructure. a,b, Coupled scans ω - 2θ of the Ge/SiP(**a**) and Ge/PSiP(70%)(**b**) heterostructures around the Si(004) and (224). **c-d,** Reciprocal space mapping of the Ge/SiP heterostructures around the symmetric Si(004) and asymmetric Si(224) reflections. **e-f,** Reciprocal space mapping of the Ge/PSiP heterostructures around the symmetric Si(004) and asymmetric Si(224) reflections. (I) relaxed Ge microcrystals, (II) asymmetric relaxation of the Ge microcrystals.”

p8 : The very low deformation field suggest that nearly no strain relaxation is observed from the side of structure.

I guess it should be written no strain is observed

We thank the reviewer for these comments.

Accordingly, we have modified the sentence as follow:

“The very low deformation field suggest **that no strain relaxation** is observed from the side of structure.”

REVIEWER COMMENTS

Reviewer #2 (Remarks to the Author):

The authors have revised the manuscript along all reviewers' comments reasonably well. No intentionally missing point. More quantitative analysis and experimental details should be valuable for other researchers.

Reviewer #3 (Remarks to the Author):

Comments have been adressed for publication.

REVIEWERS' COMMENTS

Reviewer #2 (Remarks to the Author):

The authors have revised the manuscript along all reviewers' comments reasonably well. No intentionally missing point. More quantitative analysis and experimental details should be valuable for other researchers.

We thank the reviewer for the appreciation of our work

Reviewer #3 (Remarks to the Author):

Comments have been adressed for publication.

We thank the reviewer for the appreciation of our work